# Anthelmintic Efficacy in Sheep and Goats under Different Management and Deworming Systems in the Region of Lisbon and Tagus Valley, Portugal

**DOI:** 10.3390/pathogens11121457

**Published:** 2022-12-01

**Authors:** Maria Inês Antunes, Miguel Saraiva Lima, George Stilwell, Maria Inês Romeiras, Luís Fragoso, Luís Manuel Madeira de Carvalho

**Affiliations:** 1CIISA—Centre for Interdisciplinary Research in Animal Health, Faculty of Veterinary Medicine, University of Lisbon, 1300-477 Lisbon, Portugal; 2Associate Laboratory for Animal and Veterinary Sciences (AL4AnimalS), 1300-477 Lisbon, Portugal; 3VetEquilíbrio, Lda, 2135-281 Samora Correia, Portugal

**Keywords:** gastrointestinal parasites, small ruminants, anthelmintic efficacy, FECRT, Lisbon and Tagus Valley, Portugal

## Abstract

Infections caused by gastrointestinal parasites have been described worldwide as one of the most important issues impacting small ruminant production. The systematic administration of anthelmintic (AH) drugs without following good practice principles has led to an increase in anthelmintic resistance (AR). There is scarce information regarding AH efficacy in small ruminants in Portugal. This study aimed to characterize by in vivo methods the presence and level of AR in four farms in the region of Lisbon and Tagus Valley. All four farms kept small herds in extensive management systems and used different deworming protocols. The active substances used were fenbendazole and a combination of mebendazole plus closantel in a dosage according to the manufacturers’ instructions. On day 0 (T0), fecal samples were collected from all study animals, and animals in the treatment group (n = 40) were dewormed with the AH previously chosen by the assistant veterinarian. Animals in the control group (n = 30) did not receive any AH drug. The fecal sample collection was repeated on day 15 (T15), and the control group was treated. Egg counts were performed using the McMaster method for the eggs per gram (EPG), and AR was evaluated by the fecal egg count reduction test (FECRT) with a 95% confidence level (CL). The results from this experiment indicated that the four farms presented AR with two farms to fenbendazole (FECRT of 48% and 85%) and two farms to mebendazole plus closantel (FECRT of 66% and 79%). These results indicate that the gastrointestinal parasites of the four studied Portuguese farms are resistant to benzimidazoles, which suggests an increase in AR regarding nematodes in small ruminant production systems in Portugal.

## 1. Introduction

Small ruminant production holds an important role in the Portuguese economy especially in the rural regions, where it helps fight desertification and promotes agricultural activities within the local communities. Sheep and goats are able to thrive even in more arid and less fertile regions; thus, they are preferred to large ruminants. Furthermore, the extensive management system allows for more diversified nutrition and contributes to the prevention of forest fires and soil erosion. In Portugal, sheep are reared for meat, milk and wool. Lambs and kids originating from dairy and meat farms are sold for human consumption, and the milk from dairy farms is mainly used for cheese production. However, small ruminant production is declining, mostly due to the low profitability, mandatory sanitary rules and diminished number of young farmers interested in this field of animal production [1].

Helminthic infections are a major issue, responsible for reducing both productive and reproductive performance in small ruminants [2]. To control them, farmers often administer AH drugs to their animals at frequent intervals, most of the time without following good practices, such as the timing and frequency of deworming or the selection of molecules tailored to the parasite population, type of sheep/goat production system and season. This fact has led to an increase in multi-drug resistance (MDR) in populations of gastrointestinal nematodes [3].

Nowadays, there are three major groups of AH drugs used in small ruminants: (a) benzimidazoles, such as fenbendazole, albendazole and mebendazole; (b) macrocyclic lactones, such as ivermectin, eprinomectin and moxidectin; and (c) imidazothiazoles, such as levamisole. There is also the small group of the salicylanilides and substituted phenols, in which closantel is included [4,5].

The development of AR against AH classes, such as benzimidazoles and macrocyclic lactones, has been reported all over the world. Efficacy studies carried out in the USA, Brazil, Africa, Australia, New Zealand and Europe have identified substantial gastrointestinal nematode (GIN) resistant strains [4,6]. In Europe, multiple resistance to the three major AH classes was described in sheep flocks in Scotland [7,8]. Bosco et al. [9] performed a study in 10 sheep farms in Italy, observing high AH efficacy for albendazole and ivermectin in eight farms, “normal” efficacy for macrocyclic lactones in two farms, “reduced” efficacy for albendazole in one farm and “suspected” efficacy in another farm. Furgasa et al. [10] described the presence of resistant strains of GIN against albendazole and ivermectin in a study performed in sheep at the Haramaya University farms (Ethiopia). In India, AR in goats was reported for fenbendazole and suspected for ivermectin [11].

Due to the barely known AR level in small ruminants in Portugal, the present study aimed to evaluate the AH efficacy of deworming sheep and goats under different management and deworming systems without interfering with the usual animal health program designed by the assistant veterinarians.

## 2. Materials and Methods

### 2.1. Geographic Location of the Farms and Time of the Visits

This study was performed between September 2018 and January 2020 in four small-ruminant farms (L, Q, R and M) located in the district of Santarém, region of Lisbon and Tagus Valley.

### 2.2. Management and Prior Deworming Protocol

The four farms belonged to small holders with few animals (14 to 22) reared for meat in an extensive production system. Farms L and Q had both sheep and goats, while farms R and M had only sheep. These animals were usually dewormed every six months with fenbendazole or a combination of mebendazole and closantel, so the time of the study was selected in accordance with the subsequent required deworming designed by the assistant veterinarian. 

### 2.3. Anthelmintics Used in the Experimental Study

In the mixed farms L and Q, fenbendazole (Panacur^®^ 2.5%. MSD Animal Health, Ltd., Oeiras, Portugal) was the chosen molecule and administered orally at a dosage of 5 mg/kg body weight (b.w.) according to the manufacturer’s instructions. For the sheep in farms R and M, mebendazole in association with closantel (Seponver^®^ PLUS (75 mg + 50 mg, Ecuphar Veterinaria S.L.U, Barcelona, Spain) was given orally at a dosage of 15 mg/kg b.w. according to the manufacturer’s instructions. All these dewormers were normally used at the same dosage by the farm practitioner.

### 2.4. Target Animals

The sheep were essentially White Merino and Île de France crossbreeds, and the goats were also crossbreeds raised for meat. The animals’ ages ranged from a minimum of six months to a maximum of nine years, since the farmers kept some females for breeding. All animals were included in the study, and each sample corresponded to an animal that was identified and analyzed individually. Animals to be included in the treatment (n = 40) or control groups (n = 30) were selected randomly, with age and sex being evenly distributed between both groups. Table 1 shows the distribution of samples per species collected by farm in order to perform the AH efficacy study, and Table 2 shows the species, production system and the AH used per farm.

### 2.5. Protocol

On day 0 (T0), fecal samples were collected from all the animals from both treatment and control groups. The animals from the treatment group were then dewormed with the AH previously selected. On day 15 (T15), the fecal sample collection was repeated, and the control group was also dewormed [12,13]. All animals in the treatment group had fecal egg counts (FEC) above 200. For the present study, a total of 140 samples were collected, identified and posteriorly analyzed at the laboratory. Table 3 shows the minimum and maximum value of EPG in each group in T0 and T15.

### 2.6. Sample Collection and Storage

The sample collection was performed directly from the animal while defecating or from the ground soon after defecation and exceptionally from the animal’s rectum using plastic bags, gloves and lubricant when needed. The samples were identified, stored in a cooling container and transported to the laboratory’s refrigerator, where they were maintained at a temperature of 4 °C and posteriorly analyzed within the following 48 h.

### 2.7. Laboratory Work

The fecal samples were analyzed in the Parasitology and Parasitic Diseases Laboratory of the Center for Interdisciplinary Research in Animal Health (CIISA-FMV-ULisboa). The egg shedding level was determined by counting the EPG using the McMaster slide chamber technique, and fecal cultures were performed in order to assess the most prevalent/abundant genera of gastrointestinal strongyles [14,15,16]. The laboratory work was performed during the field work period, meaning it was developed between September 2018 and January 2020, every time a sample analysis was required. 

### 2.8. Data Analysis

Data was stored, organized and statistically analyzed in Microsoft^®^ Office Excel for Mac version 16.33. The effectiveness of the AH was evaluated in an Excel spreadsheet created by Angus Cameron (AusVet Animal Health Services for the University of Sidney). The calculations were based on those of the RESO FECRT analysis program version 2, by Leo Wursthorn and Paul Martion of CSIRO, Animal Health Laboratory; these calculations are based on those published in 1989 by CSIRO ‘Anthelmintic Resistance’: Report of the Working Party for the Animal Health Committee of the SCA. The anthelmintic efficacy was evaluated by the fecal egg count reduction test (FECRT) according to Coles et al. [17,18], where the percentage reduction in egg counting is calculated by the formula:FECR = 100 (1 − X_T_/X_C_) (1)

Following this calculation, if the percentage reduction in egg count was less than 95% and the lower limit of 95% confidence level was less than 90%, resistance was present. 

## 3. Results

In the present study, all four farms presented AR.

For Farm L (sheep and goats) where the AH chosen was fenbendazole, the FECRT showed a percentage reduction of only 48% in the treatment group, which means that AR is present. In Table 4, the calculation for drench effectiveness on Farm L is shown, and Table 5 shows the summary results for the most prevalent genera identified. In this farm, in addition to *Haemonchus contortus* and *Trichostrongylus colubriformis*, there were also *Oesophagostomum venulosum*, *Chabertia ovina* and *Strongyloides papillosus* L3 found after coprocultures.

In Farm Q (sheep and goats) where the AH chosen was also fenbendazole, the FECRT showed a percentage reduction of 84%, which still means that AR is present. In Table 6, the calculation for drench effectiveness on Farm Q is shown, and Table 7 presents the most prevalent genera identified. In this farm, in addition to *Haemonchus contortus* and *Trichostrongylus colubriformis*, there were also *Chabertia ovina* and *Strongyloides papillosus* L3 found after coprocultures.

In Farm R where the AH chosen was mebendazole plus closantel, the FECRT showed a percentage reduction of 66%, which means that AR is present. In Table 8, the calculation for drench effectiveness on Farm R is shown, and Table 9 summarizes the results regarding the most prevalent genera. In this farm, in addition to *Haemonchus contortus* and *Trichostrongylus colubriformis*, there were also *Chabertia ovina* and *Strongyloides papillosus* L3 found after coprocultures.

In Farm M where the AH chosen was also mebendazole plus closantel, the FECRT showed a percentage reduction of 79%, which means that AR is present. In Table 10, the calculation for drench effectiveness on Farm M is shown, and Table 11 presents the results for the most prevalent genera. In this farm, in addition to *Haemonchus contortus* and *Trichostrongylus colubriformis*, there were also *Oesophagostomum venulosum, Chabertia ovina, Bunostomum* sp. and *Strongyloides papilosus* L3 found after coprocultures. *Haemonchus contortus* and *Trichostrongylus colubriformis* were not found in the second fecal culture of the treated group, suggesting they were susceptible to the treatment.

In Table 12, a brief summary of the target species, percentage reduction and, consequently, drench effectiveness according to the Excel spreadsheet created by Angus Cameron is presented.

## 4. Discussion

Following the objective of this study to characterize the presence and level of AR in four farms in the region of Lisbon and Tagus Valley, all the farms presented a reduced AH efficacy, evaluated by the FECRT according to Coles et al. [17,18]. In farms L, Q, R and M, the FECRT presented a reduction of 48%, 84%, 66% and 79%, respectively. In farms L and Q (sheep and goats), the drench used was fenbendazole, and the study revealed the lack of efficacy for this AH. Resistance to this molecule was also found by Sudan et al. [11] in the treated group of 10 goats in India, with a percentage reduction of 71%. Nonetheless, Tramboo et al. [19] reported fenbendazole’s efficacy at 99% in a group of 30 sheep in Kashmir Valley (India). These results may suggest that AR is more prevalent in goats, probably due to the fact that this species is usually underdosed, promoting the selection of resistant strains [20]. In farms R and M (sheep flocks), the drench used was a combination of mebendazole and closantel, and this study also revealed the presence of resistance to this AH. Even though Tramboo et al. [19] reported an efficacy of 98% for closantel in a group of 30 sheep in India, Furgasa et al. [10] showed the development of resistance against albendazole, which is structurally related to mebendazole, by GIN in sheep in Haramaya University (Ethiopia). Sargison et al. [8] reported that the field population of parasitic nematodes from a sheep flock in southeast Scotland was resistant to benzimidazoles, imidazothiazoles and both ivermectin and moxidectin macrocyclic lactone anthelmintics. Regarding the study of AH efficacy in Portugal, Mateus et al. [21], in a study performed in Northern Portugal, revealed that even though the EPG level that sheep were excreting (below 100) did not require deworming actions, AR was already present, especially to benzimidazoles. Anastácio [22] reported a “doubtful” efficacy of netobimin and resistance to diclazuril in GIN and *Eimeria* spp. in lambs, since the FECRT resulted in percentage reductions of 91.75% and 55.38%, respectively.

Despite the limitations of this study, such as the size of the farms and the requirement to maintain the dosage of AH for both species as described by the manufacturers’ instructions, these results demonstrate for the first time, under normal management conditions, the presence of in vivo AR in small ruminants in Portugal. The widespread and high presence of GI parasites in all farms of this study suggest that the deworming procedures may be failing, since AR was demonstrated in the four farms, and most animals were infected with more than one GI parasite. These results are in accordance with the results from most referred studies, which unfortunately may indicate the development of MDR nematode strains in Portugal. 

## 5. Conclusions

The growing resistance of GIN in sheep and goats to several anthelmintics is becoming a serious worldwide problem in veterinary medicine. Results from the present study indicate that the GIN of the four studied Portuguese farms were resistant to benzimidazoles due to the lack of efficacy of fenbendazole and of the combination of mebendazole and closantel. Controlling the increasing AR is urgent in order to reduce significant complications in animal health and welfare. The recommendations for the farms in this study should be to reduce the frequency of the use of anthelmintics, adjust the dosage according to species and, if possible, implement alternatives, such as rotational grazing, plowing the soil and/or keeping some parasites in refugia using the targeted selective treatment approach as shown in Figure 1 [23,24].

## Figures and Tables

**Figure 1 pathogens-11-01457-f001:**
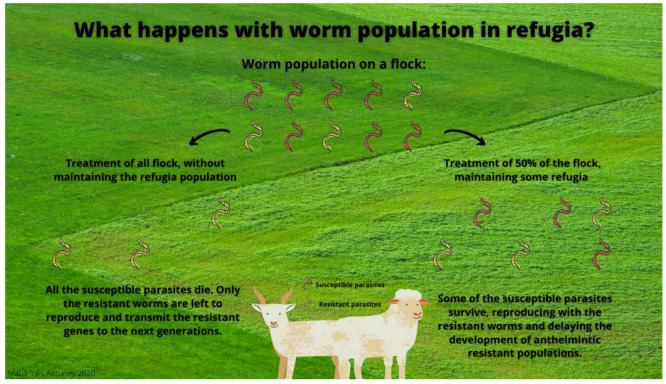
Scheme explaining how to keep refugia population (Original).

**Table 1 pathogens-11-01457-t001:** Number of animals submitted to the anthelmintic efficacy study per species and number of fecal samples collected (T0 and T15).

Farm	Treatment Group		Control Group		Total of Animals	Total of Samples
	Sheep	Goats	Sheep	Goats		
L	11	2	9	-	22	44
Q	4	4	2	4	14	28
R	11	-	9	-	20	40
M	8	-	6	-	14	28
Total	40	8	26	4	70	140

**Table 2 pathogens-11-01457-t002:** Species, production system and anthelmintic chosen per farm.

Farm	Animal Species	Production System	Anthelmintic
L	Sheep and Goats	Extensive	Fenbendazole
Q	Sheep and Goats	Extensive	Fenbendazole
R	Sheep	Extensive	Mebendazole plus closantel
M	Sheep	Extensive	Mebendazole plus closantel

**Table 3 pathogens-11-01457-t003:** Minimum and maximum number of EPG in each group in T0 and T15.

Farm	Treatment Group		Control Group		Treatment Group		Control Group	
	Min (T0)	Max (T0)	Min (T0)	Max (T0)	Min (T15)	Max (T15)	Min (T15)	Max (T15)
L	200	950	100	650	0	450	100	750
Q	200	2900	300	5800	0	1050	250	4050
R	350	15,100	350	8400	200	1650	250	10,300
M	300	21,300	400	5950	0	1500	50	6750

**Table 4 pathogens-11-01457-t004:** Drench effectiveness in Farm L.

FECRT for Farm L (Sheep and Goats)			
Drench	Pre-Test	Control	Fenbendazole
Number of animals	13	9	13
Arith. Mean	438	228	119
Var (FEC)	65,897	40,694	24,808
% Reduction			48
Var (Reduction)			0.22
Upper 95% CL			81
Lower 95% CL			−41
Drench effectiveness			Resistant

**Table 5 pathogens-11-01457-t005:** Summary results for Farm L.

Percent Fecal Egg Count Reduction (FECR)	
**Drench**	**Fenbendazole**
All species	48
Sp. *Trichostrongylus*:	−78
Sp. *Haemonchus*:	79
Sp. Other:	95
**Drench Effectiveness**	
**Drench**	**Fenbendazole**
All species	Resistant
Sp. *Trichostrongylus*:	Resistant
Sp. *Haemonchus*:	Resistant
Sp. Other:	Resistant
**Lower CL for Percent FECR**	
**Drench**	**Fenbendazole**
All species	−41
Sp. *Trichostrongylus*:	−378
Sp. *Haemonchus*:	44
Sp. Other:	86

**Table 6 pathogens-11-01457-t006:** Drench effectiveness in Farm Q.

Farm Q (Sheep and Goats)			
Drench	Pre-Test	Control	Fenbendazole
Number of animals	8	6	8
Arith. Mean	956	1675	269
Var (FEC)	837,455	1,835,750	132,813
% Reduction			84
Var (Reduction)			0.34
Upper 95% CL			95
Lower 95% CL			46
**Drench effectiveness**			Resistant

**Table 7 pathogens-11-01457-t007:** Summary results for Farm Q.

Percent Fecal Egg Count Reduction (FECR)	
**Drench**	**Fenbendazole**
All species	84
Sp. *Trichostrongylus*:	87
Sp. *Haemonchus*:	76
**Drench Effectiveness**	
**Drench**	**Fenbendazole**
All species	Resistant
Sp. *Trichostrongylus*:	Resistant
Sp. *Haemonchus*:	Resistant
**Lower CL for Percent FECR**	
**Drench**	**Fenbendazole**
All species	46
Sp. *Trichostrongylus*:	56
Sp. *Haemonchus*:	18

**Table 8 pathogens-11-01457-t008:** Drench effectiveness in Farm R.

FARM R (Sheep)			
Drench	Pre-Test	Control	Mebendazole plus Closantel
Number of animals	11	9	11
Arith. Mean	7264	2517	859
Var (FEC)	23,780,545	9,973,750	200,909
% Reduction			66
Var (Reduction)			0.20
Upper 95% CL			87
Lower 95% CL			13
Drench effectiveness			Resistant

**Table 9 pathogens-11-01457-t009:** Summary results for Farm R.

Percent Fecal Egg Count Reduction (FECR)	
**Drench**	**Mebendazole plus Closantel**
All species	66
Sp. *Trichostrongylus*:	59
Sp. *Haemonchus*:	66
Sp. Other:	77
**Drench Effectiveness**	
**Drench**	**Mebendazole plus Closantel**
All species	Resistant
Sp. *Trichostrongylus*:	Resistant
Sp. *Haemonchus*:	Resistant
Sp. Other:	Resistant
**Lower CL for Percent FECR**	
**Drench**	**Mebendazole plus Closantel**
All species	13
Sp. *Trichostrongylus*:	−5
Sp. *Haemonchus*:	13
Sp. Other:	41

**Table 10 pathogens-11-01457-t010:** Drench effectiveness in Farm M.

Farm M (Sheep)			
Drench	Pre-Test	Control	Mebendazole plus Closantel
Number of animals	8	6	8
Arith. Mean	3700	2233	475
Var (FEC)	51,883,571	8,363,667	358,571
% Reduction			79
Var (Reduction)			0.48
Upper 95% CL			95
Lower 95% CL			9
Drench effectiveness			Resistant

**Table 11 pathogens-11-01457-t011:** Summary results for Farm M.

Percent Fecal Egg Count Reduction (FECR)	
**Drench**	**Mebendazole plus Closantel**
All species	79
Sp. *Trichostrongylus*:	100
Sp. *Haemonchus*:	100
Sp. Other:	39
**Drench Effectiveness**	
**Drench**	**Mebendazole plus Closantel**
All species	Resistant
Sp. *Trichostrongylus*:	Susceptible
Sp. *Haemonchus*:	Susceptible
Sp. Other:	Resistant
**Lower CL for Percent FECR**	
**Drench**	**Mebendazole plus Closantel**
All species	9
Sp. *Trichostrongylus*:	95
Sp. *Haemonchus*:	96
Sp. Other:	−160

**Table 12 pathogens-11-01457-t012:** Summary of drench effectiveness by farm.

Farm	Drench	Species	% Reduction	Drench Effectiveness
L	Fenbendazole	Sheep and Goats	48	Resistant
Q	Fenbendazole	Sheep and Goats	84	Resistant
R	Mebendazole plus closantel	Sheep	66	Resistant
M	Mebendazole plus closantel	Sheep	79	Resistant

## Data Availability

The data presented in this study are openly available at http://hdl.handle.net/10400.5/21246 (accessed on 27 November 2022).

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
