# Peer review of "Anthelmintic Efficacy in Sheep and Goats under Different Management and Deworming Systems in the Region of Lisbon and Tagus Valley, Portugal"

_pathogens, 2022, doi:10.3390/pathogens11121457_

Round 1

Reviewer 1 Report

Dear authors,

Although your manuscript is interesting, it has regional value. Chemoresistance is well-known at all levels, including physiological mechanisms, genetic support, groups of affected substances, parasites involved, and so on. In addition, your manuscript has significant expression and structure shortcomings (see the worthless results obtained on a too-small goats study group).

Abstract section

Line 13: use "impacting" instead of "with impact on", to shorten de wording.

Line 14: you can use “increased” instead of “led to an increase”, same aforementioned reason.

Lines 15-18: divide this long sentence into two shorter and more understandable sentences, rewording also the second sentence: "There is scarce information regarding the anthelmintic efficacy in small ruminants in Portugal. So this study aimed to characterize by in vivo methods the presence and level of AR in five farms in Lisbon and Tagus Valley regions without interfering with their normal animal health program."

Lines 18-19: use "production systems and deworming protocols" instead of "types of production and deworming systems protocols". It sounds better!

Line 21: it would probably be better to use "specimens" instead of "individuals" when referring to animals.

Lines 22-23: because in English usually, the subject should precede the verb, I propose the following wording: "The fecal sample collection was repeated on day 15 (T15), and...”.

Line 27: use "The results indicate " instead of "This indicates" because it may be unclear what This refers to.

Lines 27-29: this conclusion is very general and vague. The gastrointestinal parasites might be or are resistant to benzimidazoles? I understand that your results have clearly proven AR, no? More, what does "globally resistant" mean? Globally, to all benzimidazole derivatives, or globally, meaning worldwide resistant to the benzimidazole you used? Please, clarify and reword.

Introduction section

Lines 34-36: is this sentence your statement, or was it issued by someone else and must be referenced? It looks and sounds like a statement made by someone else! More, replace "fighting" with "fight".

Lines 36-39: I reworded and rearranged your phrase as below:

"Sheep and goats have a higher capacity to obtain nutrients in more rugged lands or poor soils than larger species; furthermore, the extensive management system allows diversified nutrition and helps prevent fires and soil erosion."

Line 40: delete "purposes"! For what else, apart from milk, meat and wool, would sheep be raised?!

Line 41: "mainly", not "manly"!

Lines 45-46: I reworded your sentence as below:

“Among multiple possible causes, helminthic infections are a major issue, responsible for reducing both productive and reproductive performance in small ruminants.”

It sounds better!

Line 46: if accepting the previous comment, the sentence should begin as follows: "To control them,..." instead of "For its control, ..."

Lines 71-75: to avoid repetition of "deworming" and to simplify the statement, I reworded your sentence as below:

"Due to the barely known anthelmintic resistance in sheep from Portugal, the present study aimed to evaluate the anthelmintic efficacy of deworming in sheep and goats under different management and dehelmintization systems without interfering with the usual animal health program designed by the assistant veterinarians."

Material and Methods section

Lines 78-80: I reworded your sentence:

"This study was performed between September 2018 and January 2020 in five small ruminants farms located in the district of Santarém (Farms L, Q, R, and M) and Setúbal (Farm C)."

Lines 82, 84, and 85: use "production system" instead of "type of production".

Lines 91-100: I modified Paragraph  2.3. Anthelmintics Used in the Experimental Study, as below, to avoid repetition of "administered" and to respect the canons of the English language:

"Eprinomectin (Eprinex® Multi 5mg/ml, Boehringer Ingelheim, Animal Health, Portugal) was used in farm C since it does not require meat and milk withdrawal; the used dose rate was 0,5 mg/kg body weight (b.w.), pour-on. In the mixed-purpose farms L and Q, fenbendazole (Panacur® 2.5%. MSD Animal Health, Ltd.) was the chosen molecule, orally, at a dosage of 5 mg/kg b.w. In sheep from R and M farms, mebendazole was the used anthelmintic, in association with closantel (Seponver® PLUS (75mg + 50mg, Ecuphar Veterinaria S.L.U), given orally, at a dosage of 15 mg/kg b.w. All anthelmintics were normally used and/or chosen by the farm veterinarians."

Lines 102-103: "Sheep and goats raised in different types of management and breeds have represented target animals."

I HAVE STOPPED REVIEWING YOUR WORDING AND ENGLISH HERE. YOUR MANUSCRIPT DEFINITELY NEEDS AN EXTENSIVE SCIENTIFIC FORMULATION REVIEW! ADDITIONALLY, AN ASSESSMENT OF THE ENGLISH LANGUAGE IS RECOMMENDED.

Lines 104-105: Another example: this phrase is totally awry; you began with "The sheep from...", but further on in the sentence, you specified again "in the case of sheep". More, you twice mentioned "essentially"; this repetition is not pleasant. The modified form: "The sheep from the extensive production systems were essentially White Merino and Île de France cross-breeds raised for meat." sounds more scientific!

I WILL CONTINUE TO REFER ONLY TO THE SCIENTIFIC CONTENT.

Subsections 2.7.1. McMaster slide chamber technique and 2.7.2. Faecal cultures. I don't think describing the McMaster method or the larval cultures is necessary. They are elements well-known by the target readers! Please, remove them!

Results section

Lines 174-176: Considering your statement: "However, since there was a reduced number of animals on this trial and its faecal samples had EPG counts, the results from this calculation ended up not being significant due to the low counts" I suggest to remove all data referring to Farm C, dairy goats, from the manuscript. Keeping them in the manuscript demonstrates an insufficiency of the research. You also have the option of withdrawing the manuscript, re-doing the eprinomectin study in goats, and resubmitting it.

Also, see lines 245-249!

Lines 190-192: why do you refer to genera (e.g., Trichostrongylus spp., Oesophagostomum spp.) since fecal cultures allow species identification? Did you identify Haemonchus contortus based on those cultures but not Trichostrongylus or Oesophagostomum?

Lines 190-192, 199, 207, and 216-218: why do you refer to genera (e.g., Trichostrongylus spp., Oesophagostomum spp., Bunostomum spp.) since fecal cultures allow species identification? Did you identify Haemonchus contortus and Chabertia ovina based on those cultures but not Trichostrongylus or Oesophagostomum?

Discussion section

Lines 228-230: I'm afraid I don't understand the connection between your objective and Coles et al.! You probably wanted to refer to the corroboration of your study's data with something said by Coles et al. sometime somewhere. Still, you must clearly specify what exactly you are referring to!

Lines 237-238: I disagree with your statement: "probably due to the fact that this species is usually underdosed"! I know from personal and professional experience that slightly increased dose rates of anthelmintics are needed in goats, as the manufacturer indicates, compared to sheep. Still, under dosage does not appear due to weight evaluation; a goat rated at 50 kg based on body size, similar to a sheep of that weight, will weigh about 10 kilograms less than the sheep. This ensures the slightly increased dose required.

You did not comment at all on the occurrence mechanisms of this phenomenon and the types of chemoresistance, and you did not issue any hypotheses regarding your study. What would be the causes of the development of chemoresistance in the respective farms? Only deworming procedures failed? Other causes, such as repeated use of the same active substance on a farm, gradual increase of the dose indicated by the manufacturer, loss of the product at the time of administration (regurgitation, incorrect insertion of the self-dosing drenching gun in the mouth, etc.) and others, were not involved?.

Conclusion section

Lines 268-269: Again, I disagree with your statement: "the GIN of four of the five studied Portuguese farms might be globally resistant to benzimidazoles"!

Did you check oxfendazole (Synanthic, Fort Dodge, Mexico)? If not, "globally" is not justified in your wording.

See: Gonzalez AE, Codd EE, Horton J, Garcia HH, Gilman RH. Oxfendazole: a promising agent for the treatment and control of helminth infections in humans. Expert Rev Anti Infect Ther. 2019 Jan;17(1):51-56.

Conclusions stated between lines 271-282 are not justified; the manuscript is not a student course! Please, remove them!

Regarding figure 2 and the "other strategies" mentioned between 276 and 282 lines, I have a suggestion: the plowing of the pasture! It is probably more efficient than keeping parasites in refugia, introducing nematophagous fungal spores, formulation of vaccines, using plants with anthelmintic properties, and selective breeding of animals that tolerate nematodes, and so on! By plowing and turning the furrow, all the infective larvae and parasites' eggs will disappear deep into the soil!

Author Response

Good afternoon,

Thank you very much for your corrections and recommendations. I'm sorry for the English. I have removed the eprinomectin study and made all the corrections you've suggested. About the increased dose rates for goats, in the leaflet of our dewormer Panacur, the manufacturer does not make a distinction between the dosage for sheep and goats, that's why we used the same to be in accordance.

Regards,

Maria Inês

Reviewer 2 Report

Understanding the level of AR in small ruminants is essential to develop better parasite control programs. However, conducting these studies poses  challenges. A major issue in this research is the dose used in goats. The fenbendazole dose should have been much higher. Hence, any assessment with this drug in goats is not really valid. Also, it is unclear how animals were selected on each farm (randomly?) and if the treated and control groups were similar in age and sex. These details and a discussion regarding the correct dose for goats are needed.

The manuscript could use a review of English. Comments regarding the abstract and introduction are below.

Abstract

While word count is limited, there should be some indication of the total number of sheep/goats treated and used as controls. Also, either a min/max epg or a 95% CI for efficacy would be useful.

In several locations of the abstract abbreviations are indicated but then never used. For example, AR and EPG.

Line 119: maybe change association to combination

Line 23: with the McMaster or using the McMaster

Section 1

Line 35: either helps in fighting or helps fight

Line 39: helps prevent or helps in preventing

Line 42: However, small ruminant production

Line 61: change nematodes to nematode

Line 69: this sentence is a little awkward. Maybe, "In India, GIN anthelmintic resistance in goats was reported for fenbendazole and suspected for ivermectin."

Line 72: aimed to evaluate the anthelmintic

It might be useful to have a table with the following columns: Farm, animal species, production system, anthelmintic

Author Response

Good afternoon,

Thank you very much for your corrections and recommendations. I've made all the corrections you've suggested. About the correct and increased dose rates for goats, in the leaflet of our dewormer Panacur, the manufacturer does not make a distinction between the dosage for sheep and goats, that's why we used the same to be in accordance.

Regards,

Maria Inês

Round 2

Reviewer 1 Report

Dear authors,

I advise you (as a minor revision) to do just a short recheck because you have two 2.7 subsections :). 

2.7. Laboratory work

2.7. Data analysis

On the other hand, you made all the suggested changes, eliminated the part with eprinomectin, and improved the quality of the English language.

Author Response

Dear Reviewer,

Thank you for your corrections and kind comments. I revised the article and made some slight changes.

Best regards,

Maria Inês

Reviewer 2 Report

Thank you for the opportunity to review this manuscript. The authors have addressed many of the previous comments but there are a few that still require attention.

While the species (sheep, goat) are shown for the farms, it is not indicated how many were in the treated vs the control group. Given that the dose used for goats for fenbendazole is incorrect based on numerous publications, it needs to be clear how these were allocated or consider removing the goats from the data altogether. For some farms, there are insufficient animals to assess a FECRT. In addition, any animal with a pretreatment FEC below 200 should be excluded and the min/max of the FEC should be included in the tables. Please review the 2022 WAAVP guidelines and also consider indicating that the study was performed prior to these. While I do not doubt that there is resistance, the issues with the study design and the cautions that should be considered in interpreting the results should be indicated in the discussion at a minimum. The conclusion also needs to be more general due to the study design limitations.

Table 1. The total for sheep and goats is incorrect

Author Response

Dear Reviewer,

Thank you for your corrections and comments. I revised the article and made some changes:
- Made a table with the number of sheep and goats in each group and farm and a table with the min/max of FEC as well.
- As for the dosage of fenbendazole and the number of animals to assess a FECRT, thank you for the advice, I've added more information to make it more clear of our limitations. In fact, the farms were small but all the animals entered the study (all had more than 200 epg). As for the dose of fenbendazole, we had to use the dosage as described by the manufacturer (it does not distinguish between sheep and goats) and as the farm veterinarian used to.
- The study was performed between September 2019 and January 2020 so we used the prior WAAP guidelines.
- I've also made some changes in the discussion/conclusion in order to justify the limitations of the study as you reported.
Thank you so much for the attention. 
Best regards,

Maria Inês